# Long-Term Outcomes of Robotic-Assisted, Video-Assisted and Open Surgery in Non-Small Cell Lung Cancer: A Matched Analysis

**DOI:** 10.3390/jcm11123363

**Published:** 2022-06-11

**Authors:** Monica Casiraghi, Alessio Vincenzo Mariolo, Shehab Mohamed, Giulia Sedda, Patrick Maisonneuve, Antonio Mazzella, Giorgio Lo Iacono, Francesco Petrella, Lorenzo Spaggiari

**Affiliations:** 1Department of Thoracic Surgery, IEO—European Institute of Oncology IRCCS, 20141 Milan, Italy; shehab.mohammed@ieo.it (S.M.); giulia.sedda@ieo.it (G.S.); antonio.mazzella@ieo.it (A.M.); giorgio.loiacono@ieo.it (G.L.I.); francesco.petrella@ieo.it (F.P.); lorenzo.spaggiari@ieo.it (L.S.); 2Department of Oncology and Hemato-Oncology, University of Milan, 20122 Milan, Italy; 3Thoracic Surgery Department, Institut du Thorax Curie-Montsouris, Institut Mutualiste Montsouris, 75014 Paris, France; alessio.mariolo@gmail.com; 4Division of Epidemiology and Biostatistics, IEO—European Institute of Oncology IRCCS, 20141 Milan, Italy; patrick.maisonneuve@ieo.it

**Keywords:** stage I NSCLC, minimally invasive surgery robotic surgery, VATS

## Abstract

Introduction: This study makes a comparison between stage I non-small cell lung cancer (NSCLC) patients subjected to either robotic-assisted thoracic surgery (RATS), video-assisted thoracic surgery (VATS) or open thoracotomy, with the aim to evaluate differences between these three approaches in terms of oncological outcomes. Method: We reviewed data from 1367 consecutive patients who, between 2011 and 2017, underwent lobectomy for NSCLC with either open surgery, VATS or RATS, and performed a matched case-control study based on patients’ age, gender, clinical stage (IA, IB) and ASA score. Results: 180 patients (n = 72 RATS, n = 36 VATS, n = 72 open) were analyzed. Complication rates were found to be comparable (72.2% open, 86.1% VATS, 81.9% RATS), with similar grades of severity. The median number of resected lymph nodes was higher in open surgery (n = 22) than in VATS (n = 15; *p* = 0.0001) and in RATS (n = 17; *p* = 0.004). Pathological N2 upstaging was higher in open surgery (9.7%) compared to VATS (5.6%) and RATS (5.6%). However, the recurrence rate in VATS was significantly higher than in RATS (log rank *p* = 0.03). No statistically significant differences were detected in 5-year OS and cancer-specific survival. Conclusions: no differences were found in OS and cancer-specific survival between VATS, RATS and open lobectomy for stage I NSCLC patients; even if in VATS, the incidence of recurrences, in particular local recurrences, was higher than in RATS and in open surgery.

## 1. Introduction

Non-small-cell lung cancer (NSCLC) is a major cause of cancer death. Its poor 5-year survival rate is mostly due to the fact that usually only 20% of NSCLC patients are diagnosed when the disease is still potentially resectable and curable [1]. In the last two decades, the diffusion of lung cancer screening programs, which have reduced mortality in high-risk individuals through early detection, has led to a great increase in the number of small nodules and, therefore, of small incisions that surgeons have to deal with and perform. Consequently, the field of thoracic surgery has seen the rapid development of minimally invasive techniques, such as video-assisted thoracoscopic surgery (VATS) and robotic-assisted thoracic surgery (RATS), which have nowadays become the gold standard procedures for treating early-stage NSCLC cancer [2,3,4,5]. Indeed, compared to open surgery lobectomy, VATS and RATS are less invasive and present a series of advantages that include reduced post-operative pain and hospitalization, better functional results and improved postoperative quality of life. Despite their widely-accepted benefits, until recently, VATS and RATS had not been embraced by the community of thoracic surgeons as effective procedures. As these minimally-invasive procedures were seen to be more technically challenging and less oncologically suitable in terms of acceptable outcomes and long-term survival, open surgery and major pulmonary resection were for a long time preferred for treating lung cancer.

Even if the use of minimally invasive surgery is nowadays well-established and closed chest surgery is strongly recommended for the treatment of early-stage lung cancer [2,3,4,5,6], comparative studies between VATS, RATS and open surgery are still limited in number and provide mixed and conflicting results. To date, small retrospective single-institution studies comparing VATS with either RATS or open surgery lobectomy [7,8,9], and a few comparison studies based on large multicentric databases [10,11,12] or meta-analysis [13,14,15] have produced poor evidence on long-term oncological outcomes, often biased by the difference in techniques used and expertise developed across the various centers. This study compares stage I NSCLC patients undergoing lobectomy with either RATS, VATS or thoracotomy, with the aim of evaluating the differences between these three techniques in terms of oncological outcomes.

## 2. Materials and Methods

The study consisted in retrospectively analyzing data from 1367 consecutive patients who, between January 2011 and December 2017, underwent lobectomy for NSCLC with either open surgery, VATS or RATS at the European Institute of Oncology (IEO) in Milan, Italy. The IEO Ethical Committee approved the study and waived the need for a specific written informed consent from patients, in accordance with the policy on clinical purposes and research studies.

Inclusion criteria were: clinical stage IA and IB NSCLC (less than 4 cm in size, N0 and no evident pleural involvement) according to the 8th edition of the TNM staging system in lung cancer; intent-to-treat lobectomy with radical lymphadenectomy, performed by RATS, VATS, or open surgery; no preoperative induction therapy; no previous history of concurrent malignant disease or other previous primary lung cancer. We excluded from the study all patients with one or more of the following: histology other than NSCLC; incomplete preoperative staging; incomplete lymphadenectomy; anatomical resection other than lobectomy.

Our primary goal was the analysis of the following oncological outcomes: overall survival (OS), cancer-specific survival and disease-free survival (DFS), recurrence rate and lymph node upstaging. We also investigated conversion rate, 30-day post-operative complications, length of pleural drainage and hospital stay in order to compare short-term clinical outcomes.

### 2.1. Preoperative Staging

Preoperative staging was based on a total body computed tomography scan (CT-scan), positron emission tomography (PET) with fluorodeoxyglucose (FDG), cardiological examination, and pulmonary function test followed by anesthesia evaluation. Whenever possible, suspected mediastinal lymph node involvement was verified with endobronchial ultrasound-guided transbronchial needle aspiration (EBUS-TBNA) or mediastinoscopy. Staging and functional exams were always performed within 5 weeks before surgery. During the multidisciplinary meetings, thoracic surgeons and oncologists confirmed patient resectability and medical treatment plans.

### 2.2. Surgical Approach

All patients underwent pulmonary lobectomy and radical lymphadenectomy. In all patients, systematic lymph node dissection was performed according to the classification of the American Thoracic Society by removing all lymphatic tissue from stations 2R, 4R, 7 and 10R for right-sided tumors and from stations 5, 6, 7 and 10 L for left-sided tumors. Lesions without a preoperative diagnosis were excised by wedge resection followed by intraoperative frozen section examination.

Since the start of our robotic experience, we have been using the same 4-arm non-completely portal *robotic approach* as previously described [16,17,18,19]. Since 2016, we have been employing the Da Vinci Intuitive Surgical System (Inc., Sunnyvale, CA, USA) SI, X and XI robots without major modifications to the surgical approach. Briefly, the patient is positioned in a lateral decubitus, with the robot positioned at their head. The approach includes three-port incisions and a 3 cm utility thoracotomy in the V or IV intercostal space anteriorly with no rib spreading. The camera is introduced in the VII intercostal space on the midaxillary line. The robotic instruments are inserted through the utility incision anteriorly and through the two posterior ports in the VIII and VII intercostal spaces, respectively. The robot is driven over the patient’s shoulder at a 15° angle and attached to the 4 ports. The ports are standard for all lobectomies except that, on the right side, the camera port through the VII intercostal space is in the mid-axillary line, whereas on the left side this port is moved 2 cm posteriorly (compared with the right) to avoid the heart obscuring the vision of hilar structures.

For *VATS lobectomy*, a 3 cm long utility incision was made at either the IV or the V intercostal space anteriorly, with no rib spreading, as in the RATS approach. Then, a thoracoscopic port is placed in the VII/VIII intercostal space anteriorly for the camera, with another thoracoscopic port potentially placed in the VII intercostal space posteriorly as a utility port for the instruments. Each trocar port is maintained with a Soft Thoracoport to reduce the pressure on the intercostal nerve as much as possible. Ports placement is standard for all lobectomies.

*Open surgery* was performed via lateral muscle-sparing thoracotomy in either the IV or the V intercostal space. Pulmonary arteries, pulmonary veins and bronchial arteries were separated using a mechanical stapler through the thoracotomy or an alternative port placed in the VII or the VIII intercostal space on the mid axillary line, which was then used for the chest tube placement.

In all three approaches, pulmonary arteries, pulmonary veins and bronchial arteries were separated using a mechanical stapler (EndoGIA 30 vascular and 30 or 45 parenchyma) through one port or the utility incision/thoracotomy.

A total of 12 surgeons were involved in the study, all of them with more than 10 years’ experience as a thoracic surgeon. The first 25 lobectomies (VATS and RATS) for each surgeon were excluded to avoid any bias related to the learning curve. Only 4 surgeons out of 12 performed RATS, and only 3 performed VATS, whereas all 12 surgeons performed open surgery.

### 2.3. Post-Operative Management and Follow-Up

Patients were admitted to post-operative intensive care unit (ICU) in the first 24 h following surgery only if the anesthesiologist, based on the American Society of Anesthesiology Physical Classification System (ASA) score, deemed it necessary. Post-operative pain control during the hospital stay was managed with patient-controlled morphine administration supplemented with intra venous analgesia. Oral analgesia was administered as soon as possible to stop intravenous administration. Early patient mobilization with kinesitherapy was routinely pursued. Patients underwent daily physical examination, chest X-ray and blood tests until discharge (the day after chest drainage removal). Pleural drainage was removed in case of absence of air leak and if pleural effusion was equal or inferior to 250 mL.

Postoperative complications were defined according to the Clavien–Dindo classification [20].

Patients received a physical follow-up, chest X-ray and blood tests at 1 month after surgery; then, they received a physical examination plus a chest and upper abdomen CT-scan every 4 months for the first 2 years, every 6 months for the following 3 years, and annually after 5 years from surgery.

### 2.4. Statistical Analysis

To account for differences in the characteristics of the patients, we performed a matched case-control study: patients treated with VATS were individually matched with two patients treated with RATS and two patients treated with open thoracotomy for characteristics available at the time of surgery. Matching variables included age (±10 years), gender, clinical stage (Ia1, Ia2, Ia3, Ib) and ASA score. Conditional logistic regression was used to assess differences in the distribution of selected characteristics between the paired groups. Survival was calculated from the date of surgery to the date of death, or date of last contact with the patient. Time to recurrence was calculated from the date of surgery to the date of recurrence or the date of last contact with the patient. The cumulative incidence of recurrence during follow-up was plotted using the inverse of the Kaplan–Meier method, while survival curves were plotted using the Kaplan–Meier method. The log-rank test was used to assess differences in outcomes between paired groups (VATS vs. RATS, VATS vs. Open, RATS vs. Open). *p*-values were two-sided and those <0.05 were considered significant. All analyses were performed with the SAS software (version 9.4, Cary, NC, USA).

## 3. Results

From January 2011 to December 2017, a total of 850 patients underwent anatomical pulmonary resections. Minimally invasive surgery was performed in 517 cases (172 VATS and 345 RATS), whereas 332 patients underwent open surgery. A total of 30 patients (23 VATS and 7 RATS) were converted to thoracotomy, due to adherences or bleeding, and thus excluded from the analysis. Clinical characteristics of the entire study population stratified by surgical approach are shown in Appendix A.

Out of the 561 NSCLC patients that were at clinical stage I, 49 underwent VATS, 254 RATS, and 258 thoracotomies. Patients’ selection is shown in Figure 1.

We performed a matched case-control study of the entire cohort based on age (±10 years), gender, clinical stage (IA1, IA2, IA3, and IB) and ASA score: patients treated with VATS were individually matched with two patients treated with RATS and two patients treated with open thoracotomy. Thus, a total of 180 patients (n = 72 RATS, n = 36 VATS, and n = 72 open lobectomy) were selected and analyzed. The demographic, clinical, and surgical characteristics of the three matched groups are shown in Table 1.

No histological differences were detected between the matched populations. Most of the patients had adenocarcinoma (82.0%), followed by squamous cellular carcinoma (11%).

Pleural drainage was found to be quicker in the RATS population (median 4 days vs. 5 days for VATS vs. 5 for open surgery); however, the difference was not statistically significant (*p* = 0.07). Nevertheless, patients operated on with RATS were discharged earlier, with a median hospital stay of 5 days vs. 6 days for both the VATS and the open surgery group (*p* < 0.05).

Total complications rates resulted comparable (72.2% in open, 86.1% in VATS, and 81.9% in RATS; *p* = 0.28 open vs. VATS; *p* = 0.11 open vs. RATS; *p* = 0.36 VATS vs. RATS), with similar grades of severity. Patients’ outcomes are reported in Table 2.

A total of 4 (2.2%) Clavien–Dindo grade 3a–3b complications have been reported among the 180 patients matched: 3 (4.2%) post-operative pneumothorax requiring drainage positioning into the RATS group and 1 (1.4%) thoracotomy dehiscence requiring surgery into the open surgery group.

At the final pathological exam, 46 (63.9%) open surgery, 26 (72.2%) VATS and 65 (90.3%) RATS patients were pathological stage I; 19 (26.4%) open, 8 (22.2%) VATS and 3 (4.2%) RATS patients were pathological stage II; 7 (9.7%) open, 2 (5.6%) VATS and 4 (5.6%) RATS patients were stage III.

A higher cumulative number of lymph nodes was resected in open surgery with a median of 22 lymph nodes (range 7–49) vs. 15 lymph nodes (range 4–25) in VATS (*p* = 0.0001) and 17 (range 6–37) in RATS (*p* = 0.004). More lymph nodes (N2, Station 2/4/7) were removed in the open surgery group than in RATS and in VATS, without a significant impact on the number of N+. Pathological N2 upstaging was higher in open surgery (9.7%) compared to VATS (5.6%) and RATS (5.6%). Details on the lymph nodes removed are described in Table 2.

Post-operative adjuvant therapy was administered to 15 out of 20 (75%) patients with pathological N2 involvement. Eight (53%) patients had only chemotherapy (CT), mainly platinum-base plus vinorelbine (median cycles 4, range 1–6), and seven (47%) patients had CT plus mediastinal adjuvant radiotherapy (RT) (median 50 Gy, range 24–60 Gy). Five (25%) patients refused the adjuvant treatment proposed.

No operative mortality, in-hospital mortality or 90-day mortality occurred in the three groups of patients.

At the last follow-up (median range 5.5 years, Table 3), 131 patients (72.8%) were alive without evidence of disease, whereas 3 (1.7%) patients were alive with recurrence under treatment. Forty-four patients (24.4%) had died: 27 (15%) of the disease (14 open, 8 VATS and 5 RATS) and fourteen (7.8%) of other causes (e.g., other cancers or natural causes). Two patients were lost to follow-up.

A total of 32 patients (17.7%) developed recurrences of the disease (Table 3): 9 had local (5%), 11 regional (6.1%) and 14 distant metastases (7.7%). The incidence of total recurrences resulted significantly higher in VATS compared to RATS (log rank *p* = 0.03, Table 3 and Figure 2); in particular, local recurrences occurred in 11.1% of the cases in the VATS vs. only 2.8% in the open surgery and 4.8% in the RATS group.

Five-year disease recurrence rate was 22.9% for open surgery, 27.8% for VATS and 11.4% for RATS, whereas no statistically significant differences were detected in OS and cancer specific survival (Figure 2 and Figure 3). In particular, 5-year OS was 78.6% (67.0–86.5) for open surgery, 77.3% (59.7–88.0) for VATS and 87.4% (75.8–93.7) for RATS (*p* = 0.30, Figure 1). Cancer specific 5-year survival was 85.5% for open surgery, 80.0% for VATS and 95.7% for RATS (*p* = 0.41 VATS and *p* = 0.15 RATS; Table 3 and Figure 2).

## 4. Discussion

Controversy still exists about the optimal surgical approach to early-stage NSCLC. The ideal approach should minimize patient morbidity in order to enhance recovery, ensure a return to daily activities in a reasonable period of time and maximize oncologic outcomes. Minimally invasive approaches have long been established as suitable for resection of early-stage lung cancer [21,22], but no prospective randomized trials evaluating their relative efficacy and oncologic equivalence in comparison to open surgery are yet available. Indeed, comparative studies between VATS, RATS and open surgery are lacking, with most of them investigating short-term outcomes and providing poor evidence of comparable long-term oncological results.

One of the main ways to assess the effectiveness of NSCLC surgery is lymph node evaluation, consisting of the analysis of the total number of lymph nodes removed and the prevalence of nodal upstaging. The efficacy of lymph node dissection has been controversial since the introduction of minimally invasive surgery. A study by Lee [23] showed no difference in pN1 and pN2 upstaging in cT1 tumors between VATS and RATS (6.7% vs. 7.5% and 5.9% vs. 5%, respectively; *p* = 0.97), while data published by Wilson [24] showed similar upstaging in cT1 tumors, with pN1 and pN2 upstaging rates of 5.5% and 4.6% for RATS and open surgery, respectively. Based on the Danish Lung Cancer Registry database, Licht reported the outcomes of 1513 patients with clinical stage I NSCLC and found less upstaging in the VATS group than in the thoracotomy group, for both the N1 category (8.1% vs. 13.1%; *p* < 0.001) and the N2 category (3.8% vs. 11.5%; *p* < 0.001) [25]. Nevertheless, it has been shown that the analysis of large databases may suffer from bias, for example, in selecting the field of lymph node dissection or based on the individual surgeon’s skills and expertise, which may vary dramatically among different institutions. Using the Society of Thoracic Surgeons database, with a large volume of 11,531 cases, Boffa et al. [26] reported that N1 upstaging was significantly lower in the VATS group than in the thoracotomy group (6.7% vs. 9.3%; *p* < 0.001). The study demonstrated that the frequency of upstaging definitely increased with the operative volume of VATS performed, attesting no difference between VATS and open surgery (8.7%) in centers routinely performing procedures thoracoscopically. Recently, Kneuertz [27] analyzed upstaging in patients who underwent lobectomy for clinical N0/N1 NSCLC, showing that the highest overall rate of lymph node upstaging was in open lobectomy (21.8%), followed by robotic surgery (16.2%), and VATS (12.3%). Owing to the advantages of three-dimensional optics, and the flexible instrumentation, the thoroughness of lymphadenectomy has always been accounted as a potential strength of the robotic approach over VATS. Our results are controversial. The median number of resected lymph nodes, in particular, the mediastinal ones, was higher in open surgery (n = 22; *p* = 0.0001) and in RATS (n = 17; *p* = 0.004) compared to VATS (n = 15). Moreover, the nodal upstaging (cN0 β pN2) was higher in the open surgery group compared to RATS while no difference was reported between the two minimally invasive approaches. These results could be explained by the fact that pathological upstaging, even if it is related to the number of lymph nodes removed, it is also determined by the real N2 involvement, which for stage I NSCLC is directly related to the aggressiveness of the tumor. In our study, in fact, almost 30% of RATS patients came from our screening program, in which slow-growing tumors such as lepidic adenocarcinoma were predominant; this could be the reason for a lower pathological upstaging and a better survival rate for this group of patients. Although potentially biased by the way patients are retrospectively selected, nodal upstaging is anyway often considered a good method to assess oncologic radicality. However, in oncological surgery, the rates that best measure outcomes are OS, cancer-specific survival and in this case recurrence incidence, which allowed us to really understand the possible differences between the three approaches. In 2017, Yang [28] compared long-term outcomes among RATS, VATS, and open lobectomy in stage I NSCLC patients, showing that the 5-year OS for RATS, VATS and open surgery was 77.6%, 73.5% and 77.9%, respectively, without a statistically significant difference. Also in our study, results indicated that three surgical approaches were comparable in terms of OS and cancer-specific survival in patients treated for clinical stage I NSCLC, but that the cumulative recurrence rate was significantly higher in the VATS group. Our 5-year OS was 78.6% in thoracotomy, 77.3% in VATS and 87.4% in RATS with cancer-specific 5-year survival of 85.5%, 80.0% and 95.7%, respectively.

Our results are important since, to our knowledge, we were the first to compare the three different approaches looking not only at the short-term but also, and in particular, to the long-term oncological outcomes in patients with clinical stage I NSCLC, and using matching data collected from a single high-volume referral center. Indeed, only patients treated by surgeons with long experience in performing thoracoscopy or robotic surgery for oncological purposes were considered, thus minimizing the bias due to people being differently skilled at applying the same technique. Furthermore, matched pairs analysis potentially reduced heterogeneities between populations and allowed comparison between similar patients.

However, our study still presented several limitations. Firstly, although individual matching decreased selection bias among groups, the retrospective nature of the study might have still influenced the results. To include the maximum number of patients possible, we limited matching to the most important characteristics available at the time of surgery (age, sex, ASA score and clinical stage). Matching for other variables such as smoking status could have been performed but would have significantly reduced the overall number of patients with matched characteristics. Another important bias was related to the selection of the surgical approach and to the different surgeon’s expertise; for each case, the choice of one out of the three possible approaches could have been influenced by the patient’s preferences and/or by the particular expertise of their surgeon, considering that only 4 surgeons out of 12 performed RATS, and only 3 performed VATS, whereas all 12 surgeons performed open surgery. Furthermore, in the retrospective analysis, we were not able to compare the financial costs of the three approaches, which could be an issue in these times of increasing health care expenditure and given the hefty price of robotic technology. Nevertheless, advances in robotic technology and the possible use of the same device by multiple cancer units within the same hospital will plausibly decrease running costs and make the financial burden of the RATS approach less of a limiting factor.

## 5. Conclusions

In conclusion, our matched pairs analysis found no statistically significant differences in OS and cancer-specific survival between patients with clinical stage I NSCLC treated with either VATS, RATS or open lobectomy, even if the incidence of local recurrences was higher in VATS compared to RATS and open surgery. However, these data must be taken carefully given the selection bias that often affects retrospective studies.

## Figures and Tables

**Figure 1 jcm-11-03363-f001:**
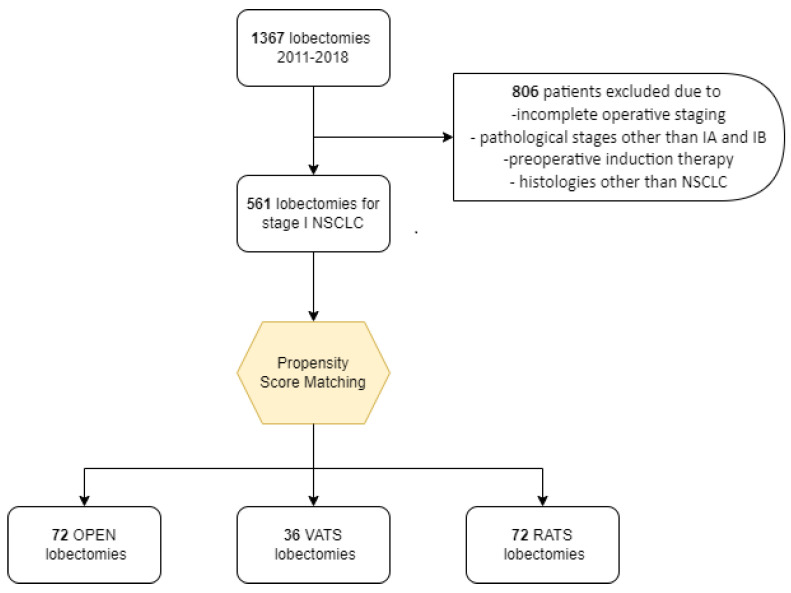
Patients’ selection algorithm.

**Figure 2 jcm-11-03363-f002:**
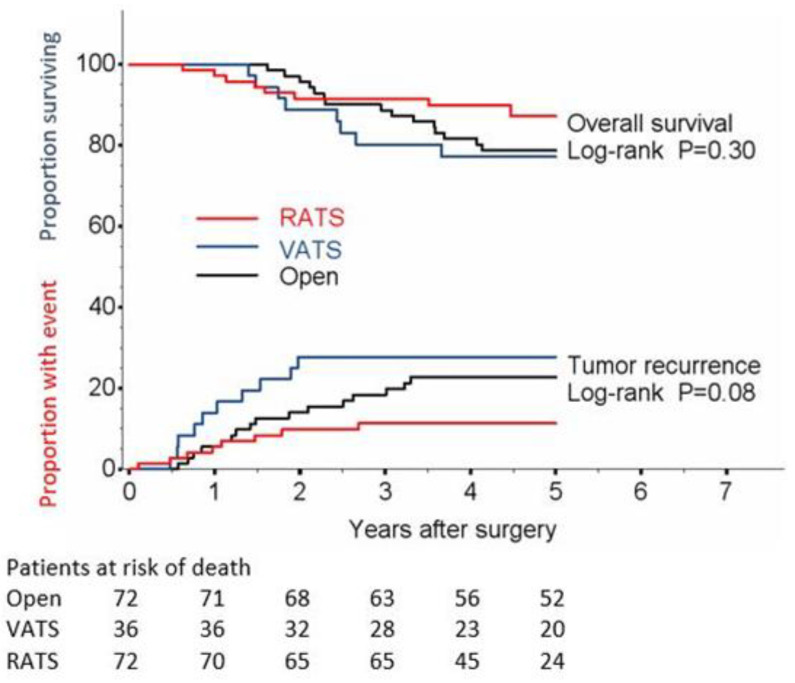
Tumor recurrence and overall survival.

**Figure 3 jcm-11-03363-f003:**
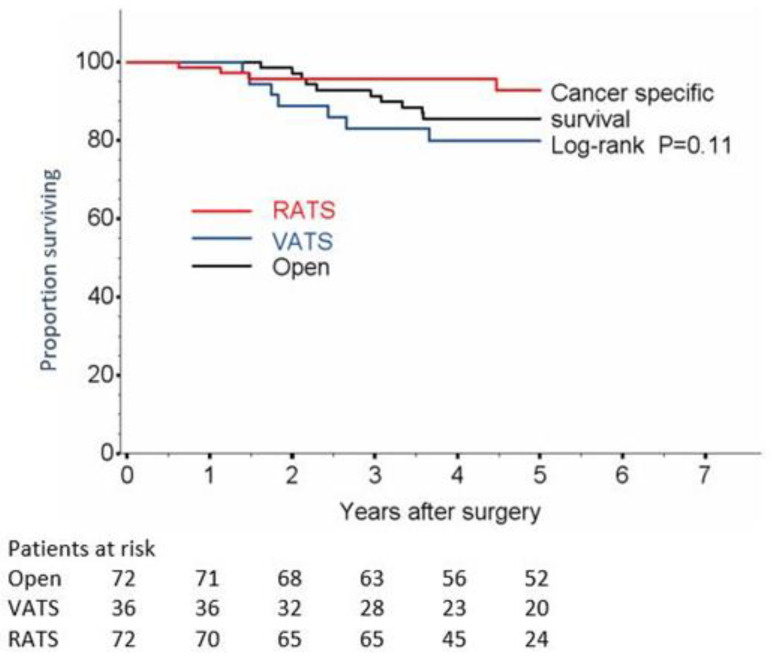
Lung Cancer-Specific survival.

**Table 1 jcm-11-03363-t001:** Patients’ characteristics.

		Open	VATS	RATS	Open vs. VATS †	Open vs. RATS †	VATS vs. RATS †
		72 * (100.0)	36 * (100.0)	72 * (100.0)			
**Age, years**, median [range]		68 (7–79)	67 (52–80)	68 (51–77)	MATCHING VARIABLE		
**Sex**	Male	32 (44.4)	16 (44.4)	32 (44.4)	MATCHING VARIABLE		
Female		40 (55.6)	20 (55.6)	40 (55.6)			
**Body mass**	Underweight	4 (5.6)	2 (5.6)	1 (1.4)			
**index**	Normal weight	38 (52.8)	15 (41.7)	29 (40.3)			
Overweight		19 (26.4)	13 (36.1)	26 (36.1)			
Obese		11 (15.3)	6 (16.7)	16 (22.2)	0.66	0.19	0.62
**Smoking status**	Never	20 (27.8)	7 (19.4)	17 (23.6)			
Former		27 (37.5)	16 (44.4)	16 (22.2)			
Current		25 (34.7)	13 (36.1)	39 (54.2)	0.58	0.06	**0.05**
**Diabetes**	No	68 (94.4)	30 (83.3)	61 (84.7)			
Yes		4 (5.6)	6 (16.7)	11 (15.3)	**0.05**	**0.05**	0.86
**Cardiac comorbidity**	No	26 (36.1)	16 (44.4)	32 (44.4)			
Yes		46 (63.9)	20 (55.6)	40 (55.6)	0.37	0.30	1.00
**Pulmonary comorbidity**	No	64 (88.9)	28 (77.8)	68 (94.4)			
Yes		8 (11.1)	8 (22.2)	(5.6)	0.12	0.23	**0.01**
**ASA score**	1	0 (0.0)	0 (0.0)	0 (0.0)	MATCHING VARIABLE		
2		66 (91.7)	33 (91.7)	66 (91.7)			
3		6 (8.3)	3 (8.3)	6 (8.3)			
**Clinical**	Stage Ia1	6 (8.3)	3 (8.3)	6 (8.3)	MATCHING VARIABLE		
**Stage**	Stage Ia2	34 (47.2)	17 (47.2)	34 (47.2)			
Stage Ia3		22 (30.6)	11 (30.6)	22 (30.6)			
Stage Ib		10 (13.9)	5 (13.9)	10 (13.9)			
**Laterality**	Left	33 (45.8)	16 (44.4)	31 (43.1)			
Right		39 (54.2)	20 (55.6)	41 (56.9)	0.89	0.74	0.88
**Site**	Upper lobe	52 (72.2)	19 (52.8)	46 (63.9)			
Medial lobe		2 (2.8)	1 (2.8)	5 (6.9)			
Lower lobe		18 (25.0)	16 (44.4)	21 (29.2)	0.17	0.38	0.22
**Histology**	Adenocarcinoma	60 (83.3)	30 (83.3)	58 (80.6)			
Squamous		9 (12.5)	4 (11.1)	7 (9.7)			
Adeno-squamous		2 (2.8)	1 (2.8)	1 (1.4)			
Other		1 (1.4)	1 (2.8)	6 (8.3)	0.96	0.24	0.71
**Diameter**, median [range]		21 [8–55]	21 [10–60]	21 [8–55]			
<10 mm		4 (5.6)	0 (0.0)	1 (1.4)			
10–19 mm		25 (34.7)	17 (47.2)	30 (41.7)			
20–29 mm		24 (33.3)	8 (22.2)	25 (34.7)			
30–49 mm		17 (23.6)	10 (27.8)	15 (20.8)			
≥50 mm		2 (2.8)	1 (2.8)	1 (1.4)	0.38	0.62	0.62
**pT**	1	51 (70.8)	26 (72.2)	49 (68.1)			
2		18 (25.0)	8 (22.2)	22 (30.6)			
3		3 (4.2)	2 (5.6)	1 (1.4)			
4		0 (0.0)	0 (0.0)	0 (0.0)	0.90	0.52	0.35
**pN**	0	50 (69.4)	29 (80.6)	66 (91.7)			
1		15 (20.8)	5 (13.9)	2 (2.8)			
2		7 (9.7)	2 (5.6)	4 (5.6)	0.41	**0.002**	0.10
**Pathological stage**	I	46 (63.9)	26 (72.2)	65 (90.3)			
**(TNM 8th edition)**	II	19 (26.4)	8 (22.2)	3 (4.2)			
III		7 (9.7)	2 (5.6)	4 (5.6)	0.56	**0.0003**	**0.01**

* Matched on age (±10 years), sex, ASA (1, 2, 3) and clinical stage (Ia1, Ia2, Ia3, Ib). † According to the univariate conditional logistic regression score test. Bold text indicates a statistically significant difference with a *p*-value less than 0.05.

**Table 2 jcm-11-03363-t002:** Patients’ outcomes.

		Open n = 72	VATS n = 36	RATS n = 72	Open vs. VATS †	Open vs. RATS †	VATS vs. RATS †
**Postoperative complications**	None	52 (72.2)	31 (86.1)	59 (81.9)			
Minor		18 (25.0)	5 (13.9)	9 (12.5)			
Major		2 (2.8)	0 (0.0)	4 (5.6)	0.28	0.11	0.36
Clavien 1		2 (2.8)	1 (2.8)	1 (1.4)			
Clavien 2		17 (23.6)	4 (11.1)	9 (12.5)			
Clavien 3a		1 (1.4)	0 (0.0)	1 (1.4)			
Clavien 3b		0 (0.0)	0 (0.0)	2 (2.8)	0.45	0.27	0.78
**Clavien 3a–3b**	Complications	1 (1.4)	0 (0.0)	3 (4.2)	0.48	0.32	0.22
**only**	Cardiac	1 (1.4)	0 (0.0)	1 (1.4)	0.48	1.00	0.49
Pulmonary		1 (1.4)	0 (0.0)	0 (0.0)	0.48	0.32	-
Surgical		0 (0.0)	0 (0.0)	2 (2.8)	-	0.16	0.32
**Lymph node**	median [range]	22 [7–49]	15 [4–25]	17 [6–37]	**0.0001**	**0.004**	**0.04**
**Removed**	Positive	0 [0–19]	0 [0–13]	0 [0–12]	0.95	0.39	0.40
Ratio Positive/Removed (%)		3% ± 10%	5% ± 16%	2% ± 10%	0.57	0.52	0.33
**N1 Removed**	median [range]	10 [1–38]	9 [1–20]	10 [1–32]	0.12	0.33	0.50
Positive		0 [0–3]	0 [0–11]	0 [0–9]	0.31	0.34	0.17
Ratio Positive/Removed (%)		5% ± 12%	5% ± 18%	3% ± 12%	0.95	0.22	0.37
**N2 Removed**	median [range]	10 [0–28]	4 [1–14]	6 [0–21]	**<0.0001**	**0.002**	**0.04**
Positive		0 [0–16]	0 [0–2]	0 [0–8]	0.44	0.55	0.58
Ratio Positive/Removed (%)		2% ± 11%	3% ± 12%	2% ± 8%	0.90	0.59	0.65

† According to the univariate conditional logistic regression score test. Bold text indicates a statistically significant difference with a *p*-value less than 0.05.

**Table 3 jcm-11-03363-t003:** Disease recurrence and overall survival.

	Open	VATS	RATS	Open vs. VATS †	Open vs. RATS †	VATS vs. RATS †
**Median follow-up**	5.0 years	5.0 years	4.4 years			
**Disease recurrence**				Log-rank	Log-rank	Log-rank
Number of events	16 (22.2)	10 (27.8)	8 (11.1)	0.45	0.09	**0.03**
Local (±regional ± distant)	2 (2.8)	4 (11.1)	3 (4.2)	0.07	0.66	0.15
Regional (±distant)	6 (8.3)	2 (5.6)	3 (4.2)	0.71	0.30	0.69
Distant only	8 (11.1)	4 (11.1)	2 (2.8)	0.89	0.05	0.07
1-year relapse, % (95% CI)	5.6 (2.1−14.2)	13.9 (6.0−30.2)	5.6 (2.1−14.2)			
2-year relapse, % (95% CI)	14.1 (7.9−24.6)	27.8 (16.0−45.5)	9.9 (4.9−19.7)			
3-year relapse, % (95% CI)	18.4 (11.1−29.6)	27.8 (16.0−45.5)	11.4 (5.9−21.5)			
4-year relapse, % (95% CI)	22.9 (14.7−34.7)	27.8 (16.0−45.5)	11.4 (5.9−21.5)			
5-year relapse, % (95% CI)	22.9 (14.7−34.7)	27.8 (16.0−45.5)	11.4 (5.9−21.5)			
HR (95% CI)	1.00 (ref)	1.35 (0.61−2.98)	0.49 (0.21−1.15)			
*p*-value		0.45	0.10			
**Cancer-specific survival (CSS)**				Log-rank	Log-rank	Log-rank
Number of deaths	10 (13.9)	7 (19.4)	4 (5.6)	0.40	0.14	0.10
1-year CSS, % (95% CI)	100	100				
2-year CSS, % (95% CI)	97.2 (89.1−99.3)	88.9 (73.1−95.7)	98.6 (90.5−99.8)			
3-year CSS, % (95% CI)	91.4 (81.9−96.0)	83.2 (66.3−92.1)	95.7 (87.3−98.6)			
4-year CSS, % (95% CI)	85.5 (74.6−91.9)	80.0 (62.4−89.9)	95.7 (87.3−98.6)			
5-year CSS, % (95% CI)	85.5 (74.6−91.9)	80.0 (62.4−89.9)	95.7 (87.3−98.6)			
HR (95% CI)	1.00 (ref)	1.50 (0.57−3.94)	0.42 (0.13−1.35)			
*p*-value		0.41	0.15			
**Overall survival (OS)**				Log-rank	Log-rank	Log-rank
Number of deaths	15 (20.8)	10 (27.8)	8 (11.1)	0.72	0.21	0.17
1-year OS, % (95% CI)	100	100	97.2 (89.3−99.3)			
2-year OS, % (95% CI)	95.8 (87.5−98.6)	88.9 (73.1−95.7)	91.6 (82.3−96.1)			
3-year OS, % (95% CI)	88.7 (78.7−94.2)	80.4 (63.3−90.2)	91.6 (82.3−96.1)			
4-year OS, % (95% CI)	81.6 (70.4−88.9)	77.3 (59.7−88.0)	90.0 (80.1−95.1)			
5-year OS, % (95% CI)	78.6 (67.0−86.5)	77.3 (59.7−88.0)	87.4 (75.8−93.7)			
HR (95% CI)	1.00 (ref)	1.16 (0.49−2.74)	0.57 (0.24−1.35)			
*p*-value		0.74	0.20			

† According to the univariate conditional logistic regression score test. Bold text indicates a statistically significant difference with a *p*-value less than 0.05.

## Data Availability

Available upon request.

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
