# Peer review of "Long-Term Outcomes of Robotic-Assisted, Video-Assisted and Open Surgery in Non-Small Cell Lung Cancer: A Matched Analysis"

_jcm, 2022, doi:10.3390/jcm11123363_

Round 1

Reviewer 1 Report

Thank you for the opportunity to review this interesting paper.

The authors conducted a study comparing the oncologic outcomes of RATS, VATS, and open thoracotomy in cStage I NSCLC patients using the PSM. They selected surgically resected 180 lung cancer patients after matching and compared the complication rate, number of dissected lymph nodes, rate of nodal upstaging, and long-term prognosis (RFS, OS) of each approach. Finally, they concluded that no differences were found in OS and cancer-specific survival between VATS, RATS and 27 open lobectomy for stage I NSCLC patients even if in VATS the incidence of recurrences, in partic-28 ular local recurrences, was higher than in RATS and in open surgery.

Although PSM was performed in this retrospective study, it did not fully correct for treatment selection bias (e.g., pStage remained significantly different between counties even after matching), and the conclusions drawn are not necessarily appropriate. The authors should mention this point and should clearly describe the criteria for selecting each approach.

The authors mentioned significantly poorer RFS with VATS, what differences in recurrence types were observed with each approach? At the risk of sounding harsh, could this be a result of the immaturity of the VATS technique at the authors' institution?

Author Response

REVIEWER 1

Comment 1: Although PSM was performed in this retrospective study, it did not fully correct for treatment selection bias (e.g., pStage remained significantly different between counties even after matching), and the conclusions drawn are not necessarily appropriate. The authors should mention this point and should clearly describe the criteria for selecting each approach.

Answer 1: We did not used propensity score matching (PSM) as indicated in the old title, but 2:1:2 individual matching based on variables available before surgery (age ±10 years, sex, ASA (1, 2, 3) and clinical stage (Ia1, Ia2, Ia3, Ib). Pathological stage takes into account the number of positive lymph nodes, which may depend on the type of surgery. We now clearly mentioned this point in the statistical methods section and discussed it later in the manuscript.

Comment 2: The authors mentioned significantly poorer RFS with VATS, what differences in recurrence types were observed with each approach?

Answer 2: difference in recurrence type for each approach was specified in table 3 and into the text at page 10 line 245.

Comment 3: At the risk of sounding harsh, could this be a result of the immaturity of the VATS technique at the authors' institution?

Answer 3: A total of 12 surgeons were involved into the study, all of them with more than 10 years’ experience as thoracic surgeon. The first 25 lobectomies (VATS and RATS) for each surgeon were excluded to avoid any bias related to the learning curve. Only 4 surgeons out of 12 performed RATS, and only 3 performed VATS, whereas all 12 surgeons performed open surgery. Besides, our center has much more experience in RATS (more than 20 years) than in VATS, and this is why in our matched case-control study the patients treated with VATS were individually matched with two patients treated with RATS and two patients treated with open thoracotomy.

Reviewer 2 Report

I would like to congratulate the authors for their interesting and informative paper.

This is a single-centre, retrospective study investigating primarily the long-term oncological outcomes and secondly the short-term surgical results of three different surgical approaches, namely video-assisted thoracoscopic surgery (VATS), robotic-assisted thoracoscopic surgery (RATS), and open thoracotomy with rib spreading, for the treatment of clinical stage I non-small-cell lung cancer. The authors performed a matched case-control study based on patient age, gender, clinical stage (IA or IB), and ASA score, including 180 patients (36 VATS, 72 RATS, and 72 open).  Rates and severity of postoperative complications were similar between the three groups. The median number of resected lymph nodes was higher with open surgery than with VATS or RATS. Similarly, pathological N2 upstaging was higher in the open-surgery group compared to the VATS and RATS groups. Of note, the recurrence rate was significantly higher in the VATS group compared to the RATS group. However, no statistically significant differences were observed in 5-year cancer-specific and overall survival between VATS, RATS, and open lobectomy.

This is an overall well-written study. The title of the manuscript accurately describes the content of the paper. The introduction provides enough information to set the background even for the reader with little knowledge on the investigated topic. The methods are adequately described to allow replication of the results. The results are clearly presented with relevant tables and figures. The findings are discussed in relevance with the pertinent literature, and the conclusions drawn are based on the findings of the study.

The present study lacks novelty. However, it should be noted that outcome data comparing lobectomy via thoracotomy, VATS, or RATS remain sparse and can be challenging to interpret. No large prospective randomized series are available. Considering the above, this study may add significant value to the existing evidence regarding the optimal surgical approach for the treatment of early stage non-small-cell lung cancer.

Author Response

We thank the reviewer for the positive comments. We are aware that this is only a retrospective study and not a prospective randomized series, but at least it comes from a single center experience. Honestly, we already went too far with the use and application of minimally invasive techniques. Nowadays, it would be impossible to propose to a patient with early-stage NSCLC a thoracotomy approach. Should it be really correct for the patient? None of our patients will never accept it, and a randomized study would be not feasible.

Reviewer 3 Report

In this article, the authors performed a matched case-control study of the entire cohort based on age (±10 years), gender, clinical stage (IA1, IA2, IA3, IB) and ASA score: patients treated with VATS were individually matched with two patients treated with RATS and two patients treated with open thoracotomy. Thus, a total of 180 patients (n=72 RATS, n= 36 VATS, and n=72 open lobectomy) were selected and analyzed. Patients’ outcomes regarding surgical method are analyzed, and the overall survival, tumor recurrence, cancer-specific survival are depicted in the figure.

This study is a retrospective study based on registry, and the number of patients in each group are not large. Due to the nature of the study, there are some limitations.

I would like to recommend the following:

[Minor points]

1. When determining case matching variables, is it a better choice not to include smoking status in case matching? Tumor includes several pathologies among non-small cell lung cancer, and the different pathologies might affect the prognosis and the survival rate regarding the smoking status. Please explain.

2. Although the title says “propensity score analysis”, there is description about the propensity score matching in materials and method. Please add a description on this method in materials and method.

3. Even though it is mentioned briefly in the discussion section, it would have been better if the authors described the surgeon’s detailed information in the materials and method section since the complications and outcomes after surgery may depend on the surgeon. We are interested in how experienced were the surgeons in the institution, how many surgeons were included in the study, and which surgeon did which surgery, etc.

4. How did the surgeons in your institute determine which patients undergo which surgery? Are there criteria for this in your institution? It is mentioned in the discussion that the choice of one out of the three possible approaches could have been influenced by the patient’s preferences and/or by the particular expertise of their surgeon. This may lead to a bias.

5. The authors mentioned that a larger number of patients reported cardiovascular complications following VATS compared to open surgery and RATS. Of course there may be a trend, but when the p values are 0.54 and 0.56 with small number of patients, can it be said that cardiovascular complications after VATS are high? If so, is there any reason that VATS has higher cardiovascular complications?

6. Please mention the complication evaluation method in detail. When was the complication evaluated? How was it evaluated?

7. There were higher rates of Clavien-Dindo grade 3a-3b surgical complications in patients treated by RATS than VATS. Even if high-grade toxicities were not common and manageable, the toxicities such as Clavien-Dindo grade 3a-3b are still dangerous, and not easy to manage. Is it appropriate to call RATS a promising alternative in this aspect? Can you say that it should be considered? Please explain.

8. Did you add another adjuvant treatment after operation in pathologically advanced stage tumor? There is no mention about post-operative treatment in the article. It might affect the treatment outcomes. Please describe in detail.

9. Pathological N2 upstaging was significantly higher in open surgery group than in RATS group, and therefore pathological stage is higher in open surgery group (table 1). However, the cancer-specific survival is not significantly different between 2 groups. Does the pathological stage have no meaning? If you added adjuvant treatment, can you say that the survival rate is entirely due to the surgical method? Please explain.

10. The authors mentioned about the patients converted to thoracotomy. It would have been better if you analyzed conversion rate between VATS and RATS and added to Table 2.

Author Response

Comment 1: When determining case matching variables, is it a better choice not to include smoking status in case matching? Tumor includes several pathologies among non-small cell lung cancer, and the different pathologies might affect the prognosis and the survival rate regarding the smoking status. Please explain.

Answer 1: We performed 2:1:2 individual matching for the principal variables available prior to surgery. We realize that other variables such as smoking status could have been considered as matching variables, but the final number of patients available for the analysis would have further reduced. We explain this part into the discussion at page 13, line 331.

Comment 2: Although the title says “propensity score analysis”, there is description about the propensity score matching in materials and method. Please add a description on this method in materials and method.

Answer 2: We inadvertently used the term propensity score matched analysis in the title, but we used 2:1:2 individual matching for age, sex, ASA score and clinical stage. We corrected the title accordingly.

Comment 3: Even though it is mentioned briefly in the discussion section, it would have been better if the authors described the surgeon’s detailed information in the materials and method section since the complications and outcomes after surgery may depend on the surgeon. We are interested in how experienced were the surgeons in the institution, how many surgeons were included in the study, and which surgeon did which surgery, etc.

Answer 3: A total of 12 surgeons were involved into the study, all of them with more than 10 years’ experience as thoracic surgeon. The first 25 lobectomies (VATS and RATS) for each surgeon were excluded to avoid any bias related to the learning curve. Only 4 surgeons out of 12 performed RATS, and only 3 performed VATS, whereas all 12 surgeons performed open surgery. We add this information into the material and method section at page 3 line 127.

Comment 4: How did the surgeons in your institute determine which patients undergo which surgery? Are there criteria for this in your institution? It is mentioned in the discussion that the choice of one out of the three possible approaches could have been influenced by the patient’s preferences and/or by the particular expertise of their surgeon. This may lead to a bias.

Answer 4: we added into the discussion at page 13 line 335 the following paragraph: “Another important bias was related to the selection of the surgical approach and to the different surgeon’s expertise; for each case, the choice of one out of the three possible approaches could have been influenced by the patient’s preferences and/or by the particular expertise of their surgeon, considering that only 4 surgeons out of 12 performed RATS, and only 3 performed VATS, whereas all 12 surgeons performed open surgery”.

Comment 5: The authors mentioned that a larger number of patients reported cardiovascular complications following VATS compared to open surgery and RATS. Of course, there may be a trend, but when the p values are 0.54 and 0.56 with small number of patients, can it be said that cardiovascular complications after VATS are high? If so, is there any reason that VATS has higher cardiovascular complications?

Answer 5: We agree with this reviewer: The observed differences were far from being statistically significant. We removed the sentence from the results section.

Comment 6: Please mention the complication evaluation method in detail. When was the complication evaluated? How was it evaluated?

Answer 6: Clavien-Dindo classification are described into the material and method section at page 4 line 143, including reference (21).

Comment 7: There were higher rates of Clavien-Dindo grade 3a-3b surgical complications in patients treated by RATS than VATS. Even if high-grade toxicities were not common and manageable, the toxicities such as Clavien-Dindo grade 3a-3b are still dangerous, and not easy to manage. Is it appropriate to call RATS a promising alternative in this aspect? Can you say that it should be considered? Please explain.

Answer 7: A total of 4 (2.2%) Clavien-Dindo grade 3a-3b complications have been reported among the 180 patients matched: 3 (4.2%) post-operative pneumothorax requiring drainage positioning into the RATS group and 1 (1.4%) thoracotomy dehiscence requiring surgery into the open surgery group. We add this Clavien-Dindo grade 3a-3b description into the result section at page 7 line 211. We consider these complications manageable and routinely treated.

Comment 8: Did you add another adjuvant treatment after operation in pathologically advanced stage tumor? There is no mention about post-operative treatment in the article. It might affect the treatment outcomes. Please describe in detail.

Answer 8: we added this information into the results section at page 8 line 232: ”Postoperative adjuvant therapy was administered to 15 out of 20 (75%) patients with pathological N2 involvement. 8 (53%) patients had only chemotherapy (CT), mainly platinum-base plus vinorelbine (median cycles 4, range 1-6), and 7 (47%) patients had CT plus mediastinal adjuvant radiotherapy (RT) (median 50 Gy, range 24-60 Gy). Five (25%) patients refused the adjuvant treatment proposed”.

Comment 9: Pathological N2 upstaging was significantly higher in open surgery group than in RATS group, and therefore pathological stage is higher in open surgery group (table 1). However, the cancer-specific survival is not significantly different between 2 groups. Does the pathological stage have no meaning? If you added adjuvant treatment, can you say that the survival rate is entirely due to the surgical method? Please explain.

Answer 9: We thank the reviewer for the comment. We added this comment into the discussion section to better explain this controversial result at page 12 line 300: “These results could be explained by the fact that pathological upstaging, even if it is related to the number of lymph node removed, it is also determinate by the real N2 involvement, which for stage I NSCLC is directly related to the aggressiveness of the tumor. In our study, in fact almost 30% of RATS patients comes from our screening program, in which slow growing tumors such as lepidic adenocarcinoma were predominant; this could be the reason for a lower pathological upstaging and a better survival of this group of patients”.

Comment 10: The authors mentioned about the patients converted to thoracotomy. It would have been better if you analyzed conversion rate between VATS and RATS and added to Table 2.

Answer 10: VATS and RATS conversions are detailed at the beginning of the result section when we described the entire population (page 4, line 167). We did not include these patients in the matched group to avoid bias in the outcome analysis, considering that those patients were at the end treated with thoracotomy approach.

Round 2

Reviewer 1 Report

No comments.